behaviour/ecology/evolution

lekking, mating system, paternity, polyandry, sexual selection, territoriality

**Author for correspondence:**
Regina H. Macedo
e-mail: rhfmacedo@gmail.com

# An atypical mating system in a neotropical manakin

Milene G. Gaiotti[1], Michael S. Webster[2] and
Regina H. Macedo[1]

[1]Programa de Pós-Graduação em Ecologia, and Laboratório de Comportamento Animal, Departamento de Zoologia, Universidade de Brasília, Brasília, Distrito Federal 70910-900, Brazil
[2]Cornell Lab of Ornithology, and Department of Neurobiology and Behavior, Cornell University, Ithaca, NY 14850, USA

RHM, 0000-0003-3510-9172

Most of the diversity in the mating systems of birds and other animals comes at higher taxonomic levels, such as across orders. Although divergent selective pressures should lead to animal mating systems that diverge sharply from those of close relatives, opportunities to examine the importance of such processes are scarce. We addressed this issue using the Araripe manakin (*Antilophia bokermanni*), a species endemic to a forest enclave surrounded by xeric shrublands in Brazil. Most manakins exhibit polygynous lekking mating systems that lack territoriality but exhibit strong sexual selection. In sharp contrast, we found that male Araripe manakins defended exclusive territories, and females nested within male territories. However, territoriality and offspring paternity were dissociated: males sired only 7% of nestlings from the nests within their territories and non-territorial males sired numerous nestlings. Moreover, female polyandry was widespread, with most broods exhibiting mixed paternity. Apparently, territories in this species function differently from both lekking arenas and resource-based territories of socially monogamous species. The unexpected territoriality of Araripe manakins and its dissociation from paternity is a unique evolutionary development within the manakin clade. Collectively, our findings underscore how divergences in mating systems might evolve based on selective pressures from novel environmental contexts.

## 1. Introduction

There has been considerable interest in the factors that drive intraspecific variation in mating strategies and sex roles (especially in fish: [1,2]; reviewed in [3]), including variation in the frequency of extra-pair paternity [4–7]. Similarly, classic as well as more recent studies in behavioural ecology have mapped mating system variation across closely related species

within families onto variation in ecological conditions [8–11]. It is widely acknowledged that most variation in life-history traits among bird lineages, including mating system, corresponds to phylogenetic differences above the family level, and so appears to have developed relatively early in the ancestral history of birds [12]. However, environmental pressures can also play a role in shaping mating systems, and may result in the divergence of some species from the typical pattern associated with their clade. However, few studies have explored cases where a very small number of species have deviated from the social/mating system typical of their entire family, and show intermediacy and possible transition between mating system types, as we document here for a species in a clade of nearly exclusively lek-breeding birds. Examining species with breeding behaviours that deviate substantially from those of their closest relatives allows us to evaluate the mechanisms and selective forces that shape breeding ecology and that are potentially important in speciation patterns.

Avian mating systems range from social monogamy with varying levels of extra-pair fertilization [4], where males typically defend territories that contain valuable resources for females, to polygynous mating systems, where males obtain multiple mates via various strategies [8,13,14]. In polygynous systems, males can defend clumped resources or female groups, or can occupy 'lek' display courts devoid of resources [15,16]. Variance in male mating success, and hence the strength of sexual selection, is generally very strong in lekking compared to other mating systems [16], probably because males are not constrained by the need for paternal care and females are not constrained in their ability to choose males.

The manakins (family Pipridae), a clade of approximately 60 frugivorous bird species distributed across tropical forests in the Neotropics, exhibit nearly ubiquitous lekking polygyny with arena aggregations [17–21]. Males of many species are notorious for their plumage ornaments and dazzling displays that involve some of the most elaborate, vigorous and complex movements known among passerines [22–26]. Manakin display elements are very diverse, and Prum [27] suggested that they may have evolved by unconstrained evolutionary processes that reflect Fisherian mechanisms, i.e. they may not conform to the quality indicator framework frequently suggested to shape secondary sexual characters. Empirical studies of manakins to test different theoretical models are sorely lacking. Only a few species have benefited from in-depth descriptions of male courtship displays [19,28–31], assessment of male mating success [22,32–34], or analyses of the relationship between secondary sexual traits and female choice [35]. Although female polyandry has been verified in a couple of other lekking taxa (e.g. peacocks *Pavo cristatus* [36]; buff-breasted sandpipers *Tryngites subruficollis* [37]), it is suggested to be rare in most lekking birds [38]. The three studies to date examining this issue in manakins have found relatively low levels of polyandry: 5% of broods in blue-crowned manakins (*Lepidothrix coronata* [39]), 8.7% of broods in lance-tailed manakins (*Chiroxiphia lanceolata* [40]) and 18% of broods in wire-tailed manakins (*Pipra filicauda* [41]).

Lekking is considered an ancestral trait in the monophyletic Pipridae clade, with a single origin in the group's common ancestor, and very few deviations from lek mating have been documented for manakins [24]. One case is that of green manakins (genus *Xenopipo*), in which the sexes are largely monochromatic and males have lost lekking behaviour [42]. In one species of another genus, the helmeted manakin *Antilophia galeata*, no evidence has been found of the typical lek arena configuration. Differently from the green manakins, however, this species is highly dichromatic. The mating system of this species was tentatively described as monogamous, based on a few observations of apparent male territoriality and female nesting within territories [43]. Yet the proposed loss of true lekking behaviour in the helmeted manakin has remained untested for over 25 years, which is surprising given the evolutionary conundrum of a socially monogamous manakin [24]. A significant ornithological discovery occurred when a second *Antilophia* species was described approximately 20 years ago: the Araripe manakin (*A. bokermanni*) was discovered in a remote and secluded region in northeastern Brazil [44]. The species occupies a small (approx. 30 km²), humid forest area surrounded by the semi-arid Caatinga biome of northeastern Brazil, thus appearing to be considerably more habitat-constrained than other manakins. This newly discovered species was assumed to be territorial and monogamous, given plumage similarities to the helmeted manakin, despite the absence of genetic or behavioural studies to confirm this for either of the *Antilophia* species [45].

The possibility that the Araripe manakin could exhibit breeding behaviours very different from its close relatives led us to investigate its social and genetic mating system, which could shed light on selective pressures involved in the evolution of lekking systems overall. We focused on determining whether males defend territories that might contain resources, since data from the sister species suggested the existence of territoriality [43]. If confirmed, we also expected to find females nesting within male territories, and consequently, that male territory owners would sire many or all of the offspring produced by females on their territories (i.e. social and possibly genetic monogamy). Given

the possibility of territorial monogamy, we expected that males with better-quality territories (e.g. larger or containing key resources) would be favoured by females and achieve higher reproductive success. Although we anticipated a loss of elaborate courtship displays, we expected that female choice for healthier males or those exhibiting enhanced condition could also occur. Finally, we considered the possibility that paternal care of nestlings might have co-evolved with territoriality.

# 2. Methods

## 2.1. Study species and research site

We studied a population of Araripe manakins from 2013 to 2016. This endemic and threatened species, first described in 1998 [44], is known from a single locality, a 31 km$^2$ forested area in the Araripe plateau (7.3875° S, 40.2161° W) in Ceará State, Brazil, with a population currently estimated at only 800 individuals. The species is strongly dimorphic, with males in definitive plumage exhibiting white body plumage and a striking red crest and mantle, whereas females and males in predefinitive plumage are green. Araripe manakins consume both fruits and arthropods, although the former are the more critical component of their diets [46]. Females lay clutches of one or two eggs during the rainy season, and nesting success is relatively high (72%) for the Neotropics [47]. No previous study has been conducted on any aspect of the species' mating system or behaviour.

Research was carried out on the slopes of the 800 m high Araripe plateau, which forms the boundary between Ceará and Pernambuco states in northeastern Brazil. The region is contained within the Caatinga biome, a semi-arid region that characterizes most of northeastern Brazil. The structural geography of the plateau, which encompasses approximately 4500 km$^2$, is unique in that its upper areas function as a rainwater catch basin, providing sufficient moisture to sustain an equatorial type forest along the plateau sides, a vegetation landscape strikingly different from the xeric shrublands of the surrounding region. The mean annual rainfall in the region is about 934 mm with a mean annual temperature of 25.1°C. The water collected at the top of the plateau percolates and emerges along the slopes of the plateau in the form of some 130 water sources, around which most of the Araripe manakin populations reside [48]. Field activities were conducted in six distinct localities along the plateau's slopes, encompassing a total study area of approximately 1.27 km$^2$.

## 2.2. Field methods

Across three field seasons (2013–2016), we banded 350 adult Araripe manakins (181 males, 169 females) and 119 nestlings with metal and colour bands, allowing us to identify them individually and monitor their behaviour. Males with definitive plumage have entirely white body plumages and vivid red helmets and mantles, whereas males with predefinitive plumages have either entirely green plumages, similar to females, or exhibit some green feathers amid the white plumage. Males with definitive plumages were easily identified while molecular sexing of all individuals (details below) enabled us to distinguish green males (with predefinitive plumage) from adult females. One male offspring that was banded in the nest and recaptured 2 years later still retained the predefinitive green plumage, an observation that is in line with the delay in plumage maturation typical of other piprids. We took morphological measurements from all birds, including: weight (g), right wing, left tarsus, beak and tail lengths. For males, we also measured the height of the helmet crest and the length of the mantle. We collected blood samples (approx. 0.2 ml) from all birds via brachial venipuncture and stored them in ethanol 99% for further genetic analysis (see below).

We found nests by inspecting the vegetation and following individuals carrying nesting materials. Once found, we monitored nests ($n = 190$) every 2 days and then daily near the estimated hatching date. Of these nests, 101 became active (i.e. had eggs and/or chicks), resulting in 119 nestlings, which were measured and blood sampled. To assess the roles of males and females in parental care, we video-recorded 20 nests for a total of 195 h (in bouts of 90 min), during incubation and nestling stages (1–2 days). Camouflaged cameras (Kodak Zx1 and Multilaser DC115, zoom 10x) were positioned approximately 2 m from nests.

Males became non-territorial and joined multi-male aggregations during the non-breeding seasons. However, once the breeding seasons started, individual males sang continuously during the day and banded individuals were observed in the same specific sites, rarely intruding upon each other's areas of activity or singing close together. We mapped male territories using a GPS (Garmin e-Trex 10) to

mark the places where they sang or interacted aggressively across a minimum of 2 days and a maximum of 11 days of focal observations per male. Males typically used very few perches for singing, and we inserted the mapped coordinates (minimum of four) of singing perches in Google Maps© and then used the polygon-designing tool to define the boundaries of each male territory. We estimated the area of each minimum convex polygon with EarthPoint© [49], a plugin for Google Maps.

## 2.3. Quantifying male body condition

A body condition index (BCI) was calculated for males based upon the ratio of mass to tarsus length, such that higher values indicate individuals with a greater mass relative to skeletal size, and which can be considered as being in better condition in terms of lipid reserves. This method is regarded as a good indicator of individual health when the two variables, mass and tarsus length, are uncorrelated [50], as was the case in this study (Pearson's $r = 0.27$; $p = 0.10$).

## 2.4. Parentage analysis and sexing

We extracted DNA using a standard extraction protocol (QIAGEN®) and sexed adults and nestlings with the primer set 2550F/2718R [51]. We genotyped samples from 96 chicks from 60 nests using 15 polymorphic microsatellite markers [52–55] and ran paternity analysis in CERVUS 3.0.0 [56]. In brief, we used CERVUS to assign parentage to the most likely candidate parents under relaxed (90%) and strict (95%) levels of confidence, by calculating the likelihood ratio scores. Critical values of these scores were estimated through simulations in CERVUS with parameters that were appropriate for our study population. Paternity was only confirmed for a nestling when: (i) the logarithm of the odds (LOD) score for the trio mother–nestling–father was higher than 6.45 (i.e. more than 95% confidence by CERVUS; [56]), (ii) the confirmed mother's LOD score was higher than 4.32, and (iii) the number of mismatching alleles between nestling and father was less than 2. In the few cases of very close LOD scores ($N = 5$), the male captured at the nearest distance to the nest was considered the father. For additional details, see the electronic supplementary material (including tables S1 and S2).

## 2.5. Quantifying reproductive and pairing success

Araripe manakin clutches contain one or two eggs that are incubated for an average of 19 days ($n = 125$ nests) and chicks fledge at 16 days on average [47]. Predation rate is approximately 30% across the egg-laying and incubation periods and about 20% for the nestling stage (Mayfield method; [47]). We considered a nest to have been successful when at least one nestling fledged. Reproductive success was based upon the genetically confirmed number of offspring produced. Because male reproductive success did not vary greatly among individuals (see Results), for statistical analyses that explored associations between male reproduction and body condition, we considered breeding success as a binary variable: 0 (when a male sired no offspring) or 1 (when a male sired at least one offspring).

## 2.6. Statistical analyses

We developed three sets of generalized linear models in R (v. 3.5.3; [57]). First, we fitted a generalized linear model (GLM) to test the effect of body condition upon breeding success, pooling territorial and non-territorial males, and another GLM using only territorial males. A third GLM was modelled to examine the association between size of territories and achievement of breeding status by males. We also used a Mann–Whitney $U$-test to determine whether territories containing water differed in size from those that had no water sources.

# 3. Results

During three breeding seasons and across six field sites, we found that males, once banded, were seen singing in the same areas consistently through time. We were able to closely monitor 51 Araripe manakin territories (Season 1: $n = 20$; Season 2: $n = 18$; Season 3: $n = 13$) defended by males in definitive plumage, 48 of which were banded; the other three territories were occupied by males we were unable to band but assumed they were the same males given the consistency of singing perches. These 48 banded males were recaptured or observed in the same general region as the original

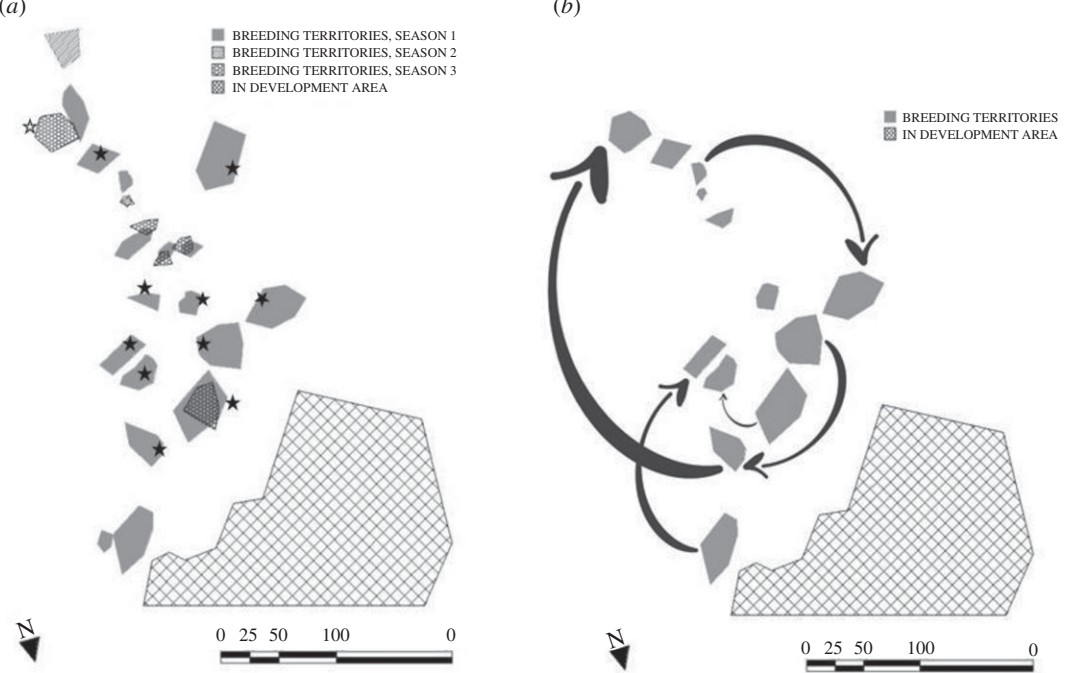

**Figure 1.** Configuration of territories and pattern of loss/gain of paternity in the Araripe manakin are consistent with the existence of territoriality and promiscuity. More intensive focal tracking of males to determine territories was conducted in Season 1, which resulted in more numerous territories demarcated for that season, as shown. In the other two seasons assessment of territoriality was a by-product of other field activities (e.g. searching for nests). (a) Twenty-four male-defended territories in two study sites in the Araripe Plateau, Ceará state, Brazil across three breeding seasons (2013–2016). The black stars show 10 territories that were held by the same males from breeding season 1 to 2, while the single white star indicates a territory held by the same male from breeding season 2 to 3; (b) Five territories in two breeding seasons (2013 and 2015) where male territory owners lost paternity within broods inside their territories (origins of arrows) while gaining paternity in broods in other males' territories (arrow points) in the Araripe Plateau, Ceará state, Brazil.

capture site an average of 8.48 times each (range 4–16 days), supporting the notion of territoriality (as functionally defined by Pitelka [58] and Schoener [59]). We estimated the size of 29 of these territories (figure 1). Estimated territory size ranged from $213\,m^2$ to $3131\,m^2$ (mean: $1209\,m^2 \pm 122$). Neighbouring territories within the study sites varied in how close they were to each other, from sharing boundaries to being 65.3 m apart. Eleven of the 29 measured territories contained water sources within their perimeters and were significantly smaller than territories without water (Mann–Whitney $U = 21$, $p = 0.01$). All 51 monitored territories were defended by males in definitive plumage; however, 11 other males in definitive plumage were never observed holding territories although they were seen frequently in the study sites, and were also captured repeatedly (at least three times) within other males' territories. By contrast, males with predefinitive plumages were not observed defending territories or singing during the breeding season.

To determine whether territoriality might be associated with female nesting and social or genetic monogamy, we monitored nests and behaviours that could be indicative of monogamy. We found 188 nests within areas occupied by singing males, 48 of which were banded. For two additional nests found, no males were ever seen in the areas. Of the 188 nests situated within male-occupied areas, monitored across the 3 years of the study, 181 areas contained a single female nest, three areas had two simultaneously nesting females, and one area had three females nesting simultaneously. These nesting patterns suggested male territoriality and social monogamy with possible low levels of polygyny.

Despite the fact that females nested within male territories, however, the genetic paternity analyses of 96 chicks from 60 nests (36 broods with two nestlings; 24 broods with one nestling) showed that this pattern was not indicative of male siring success. We identified the biological father of 57 of these chicks, and of these 53 (93%) were sired by a male other than the banded male defending the territory where the nest was located; the remaining 39 offspring did not match with any sampled male and so were apparently sired by unsampled males. The distribution of siring success across males sampled in three breeding seasons ($n = 61$) was fairly homogeneous and somewhat different from the typical high

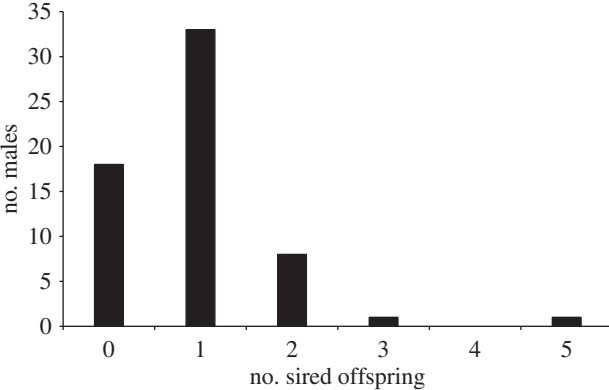

**Figure 2.** Distribution of reproductive success across three breeding seasons (2013–2016) for 61 male Araripe manakins indicates relatively little variability in reproductive success in the population in the Araripe Plateau, Ceará state, Brazil.

variance pattern seen for lekking species [60]: most sampled males sired a single offspring (figure 2). Among males that sired at least one offspring ($n = 43$), only 52% ($n = 22$) were males with definitive plumage, whereas 48% ($n = 21$) were males in predefinitive plumage, which did not defend any known territory and were never observed singing. Of 29 sampled males that defended territories, 14 (48.2%) gained paternity in at least one brood, whether within their own territory boundaries or elsewhere (figure 1). Furthermore, in 22 of 27 (81%) double-chick broods where paternity was determined, the two chicks were sired by different males. In the remaining five broods both chicks had the same father, although in none of these cases were the chicks sired by the owner of the territory where these nests occurred.

The apparent courtship behaviour of the Araripe manakin, which we were unable to objectively quantify, was fairly simple. In a half-dozen occasions, we observed two to three males conducting joint circular flights in the forest understorey (usually less than 3 m), always outside individual territories and far from any nests. The typical behaviour began with the males hopping between perches and displacing each other before initiating circular chase flights, which occurred concurrently with the production of apparently non-vocal sounds probably created mechanically by the wings. Following these flights, the males perched side-by-side, and in all cases, we recorded at least one female within sight, which suggests that the flights may function to attract or court females. As for paternal care, we verified that in 195 h of nest video recordings ($n = 20$ nests), no male ever incubated eggs or brought food to the nestlings, as compared with an average of 3.89 ($\pm$2.44) female visits/ hour/nest during nestling feeding.

When considering all Araripe males together, whether territorial or not, we found a positive relation between an index of body condition (BCI = mass/tarsus length) and achieving reproduction (i.e. production of at least one offspring) (GLM: $\chi^2 = 7.03$, $p = 0.008$; $\beta = 0.88 \pm 0.37$, $n = 43$; figure 3$a$). This positive association between body condition and reproduction remained significant when only territorial males were considered in the analysis (GLM: $\chi^2 = 5.38$, $p = 0.02$; $\beta = 1.08 \pm 0.54$, $n = 14$; figure 3$b$). In addition, males holding larger territories were somewhat more likely to produce offspring than those with smaller territories, although this relation did not quite reach statistical significance (GLM: $\chi^2 = 3.12$, $p = 0.08$; $\beta = 0.72 \pm 0.44$).

## 4. Discussion

Mating systems across a broad array of animal species tend to show most variation above the family level [12], and thus exhibit little variation at lower taxonomic levels. However, our study shows that the Araripe manakin has a mating system that differs substantially in many characteristics from that of its close relatives. Given the suggestion of a possibly monogamous mating system in a previous study of the helmeted manakin [43], in our study of the Araripe manakin we expected to find evidence of male territoriality in association with other traits typical of resource-based monogamous mating systems. Considering the earlier suggestion of social monogamy and the loss of lekking behaviour in the *Antilophia* genus [24,43], we predicted that males would have lost the elaborate courtship behaviours exhibited by other manakins, and that males would provide care for broods they sired within their territories, since most socially monogamous birds exhibit biparental care of offspring [61].

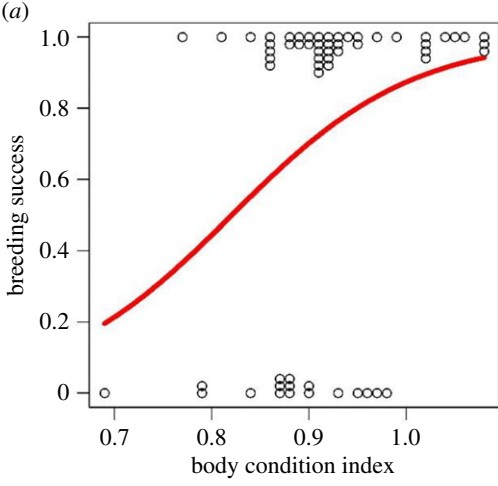

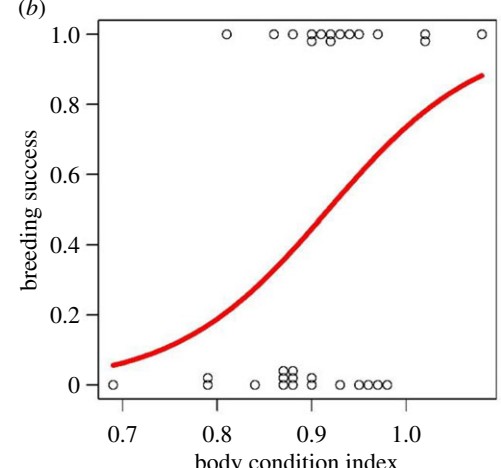

**Figure 3.** Influence of male Araripe manakin body condition on the probability of breeding success, across three breeding seasons (2013–2016). (*a*) Correlation between BCI (mass/tarsus length) and probability of offspring production (binary variable: 0 = no offspring; 1 = at least one offspring), pooling territorial and non-territorial males. The red line represents a simple logistic regression based on a GLM model. (*b*) Correlation between BCI (mass/tarsus length) and probability of offspring production (binary variable: 0 = no offspring; 1 = at least one offspring), restricted to territorial males. The red line represents a simple logistic regression based on a GLM model.

While our data confirm that territoriality occurs in the Araripe manakin, this was detached from the many other traits that typically evolve in conjunction with male resource monopolization.

Our results demonstrate that the Araripe manakin mating system is atypical for the Pipridae family, because males do not aggregate in display arenas (leks), but instead defend territories within which breeding females nest. Most unexpectedly, males sired few of the young produced on their own territories, but frequently obtained paternity elsewhere. While this indicates a high level of promiscuity, males in the population exhibited a lower degree of reproductive skew than usually seen in lekking species. Furthermore, the majority of Araripe manakin broods exhibited multiple paternity, an infrequent phenomenon in manakins or other lekking birds studied to date. Nonetheless, the species retains some traits typical of lekking manakins, such as male emancipation from offspring care and apparently clustered aerial displays, albeit less elaborate than seen in other manakins. These social aerial displays produce mechanical sounds, a common feature among other manakins [62–65], and the flight patterns we observed are similar to those described for the helmeted manakin [66]. Taken together, these data suggest that some (ancestral) lekking system elements may be partially retained in the Araripe manakin although others have been lost.

The dissociation between male territorial ownership and genetic paternity of broods is puzzling and leads to the question of why Araripe males defend territories. This question arises because territory ownership: (i) is not necessary for males to achieve at least some reproduction, and (ii) does not ensure paternity of broods within territories. Because we assessed paternity only for broods within our study sites, it is possible that females that copulated with territorial males left the study area and nested elsewhere within the limited forest habitat available. Thus, one possibility is that we underestimated the reproductive success of territorial males that, although unable to monopolize paternity within their territories, may nonetheless have attracted and copulated with more females than did non-territorial males. Females in both polygynous and socially monogamous species may favour males that are healthier, and which can provide better parental care or access to defended resources (reviewed in [67]). Given that we found a positive association between male body condition and reproduction, we speculate that such males can maintain territories containing attractive resources, perhaps fruiting vegetation or water, and may be able to achieve higher reproductive success by gaining access to more females. Additionally, it is quite possible that territories contribute towards the health and survival of the males themselves. These possibilities, however, remain to be tested.

Both Araripe and helmeted manakins occur outside of lowland tropical forest biomes typical of many manakin species, inhabiting instead geographically narrow and fragmented strips of humid forests surrounded by savannah or other xeric biomes that differ substantially from the humid forest strips [43,44,47]. It is likely that extrinsic selective factors caused the Araripe manakin to diverge substantially from the lekking system typical of its closest relatives. In fact, it is noteworthy that

*Antilophia* is a sister genus to *Chiroxiphia* [68,69], and the latter can be considered extreme within lek-mating systems because of the very high variance in mating success among males. The divergence in the mating systems of these two phylogenetically close genera strengthens the inference that ecological forces can outweigh phylogeny in shaping mating systems [8]. Among other possible mechanisms, we consider that population bottlenecks in the recent past of the Araripe manakin may have led to genetic drift in some traits. We also speculate that the historical changes in proportions of savannah and rainforest in the Amazonian basin (reviewed in [70]) may have led to disjoint distributions and isolation of some taxa in exceptionally small fragments, which may have been the case for both the Araripe and helmeted manakins. In addition to the historical constriction of forested areas occupied by the Araripe manakin, anthropogenic disturbances leading to further fragmentation in the region may also have led to changes in the species' mating system [71]. Within such constricted and fragmented patches of humid habitat, females may not be able to disperse far from male territories after mating, which probably accounts for nesting within male territories [24].

# 5. Conclusion

The non-conformity of the Araripe manakin relative to other species in the family, with a loss of typical lekking traits as well as acquisition of distinctive behavioural patterns, suggests that within this taxonomic clade there is enough plasticity to respond to environmental pressures. Studies such as this one, which demonstrate a divergence in mating behaviour relative to the ancestral condition of the clade, provide a powerful perspective to interpret the role of environmental selective factors in moulding the evolution of mating systems in general, reinforcing the classical tenet that ecology is the main driver shaping mating systems [8]. Specifically, our results suggest that selective pressures associated with novel habitat types could drive the behaviour of a species to deviate substantially from that of its closest relatives. The implications of these findings are far-reaching and relevant in current circumstances where climate change and habitat fragmentation may potentially affect selective pressures and mating patterns of birds and other animals.

Ethics. All birds were handled and banded with permission from Instituto Brasileiro do Meio Ambiente e dos Recursos Naturais Renováveis – IBAMA (permit no. 40116-4). The study was reviewed and approved by the Brazilian bird banding agency CEMAVE (licence no. 3731/2). M.G.G., a Brazilian citizen working on a thesis project, handled the birds and collected blood and feather samples in the field site in Brazil, according to the protocols established by IBAMA and CEMAVE (under the licences indicated above). At the time her thesis project was approved, no 'Animal Care Protocol' was required by the University of Brasilia.

Data accessibility. The data for this study are deposited on Dryad Digital Repository: https://doi.org/10.5061/dryad.d2547d7z5 [72].

Authors' contributions. M.G.G. and R.H.M. conceived the project. M.G.G., R.H.M. and M.S.W. developed the methodology. M.G.G. collected field data. M.G.G. and M.S.W. conducted laboratory analyses. M.G.G., R.H.M. and M.S.W. performed statistical analyses. R.H.M., M.G.G. and M.S.W. wrote the manuscript. Funding was acquired by R.H.M., M.S.W. and M.G.G. All authors gave final approval for publication.

Competing interests. The authors declare no competing interests.

Funding. Scholarships were provided to M.G.G. by Coordenação de Aperfeiçoamento de Pessoal de Nível Superior (CAPES) and Conselho Nacional de Desenvolvimento Científico e Tecnológico (CNPq). CNPq also provided a fellowship for R.H.M. Field and lab work were funded by Association of Field Ornithology, Ornithological Council, Geopark Araripe, Rufford Foundation, the Fundação de Apoio a Pesquisa do DF (FAP-DF) and the Cornell Lab of Ornithology.

Acknowledgements. We are grateful for logistic support from Universidade de Brasília and Cornell University, for field assistance provided by João H. de Oliveira, Verônica Lima, Eveny L. C. Maia and Tatyane Oliveira and for help with laboratory work provided by Bronwyn Butcher. We thank NGO Aquasis for help with access to our field study sites. We are also grateful to two anonymous reviewers who provided excellent suggestions for improving the manuscript.

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
