## [Reviewer comments · Royal Society Open Science]

Review History

RSOS-191548.R0 (Original submission)

Review form: Reviewer 1

Is the manuscript scientifically sound in its present form?

Yes

Are the interpretations and conclusions justified by the results?

Yes

Is the language acceptable?

Yes

Do you have any ethical concerns with this paper?

No

Have you any concerns about statistical analyses in this paper?

No

Recommendation?

Accept with minor revision (please list in comments)

Comments to the Author(s)

This paper describes the intriguing mating system of the Araripe manakin. Unlike more typical manakins that form leks where males have high mating skew, in this species males defend a more tradition territory and the skew in male mating success is much lower. Females of other manakins have relatively low levels of broods sired by multiple males, this may be more common in the Araripe manakin. However, unlike more traditional (socially) monogamous systems where males may sire many of the offspring raised in nests within their territory and provision offspring, males in the Araripe manakin rarely sired chicks in nests within their territories and never provisioned offspring. In addition, territory holding does not appear necessary for males to reproduce - nearly half of the offspring were sired by males that were not yet in adult plumage (and these green males were not observed defending territories). Thus this system is not quite like a typical manakin (though elements of that are present), though it is also not like a typical socially monogamous passerine.

While the overall study is well-done, and the presentation was good, there are a number of areas where I felt there should be some additional clarification or details provided to strengthen the manuscript.

Specific Comments (Page numbers refer to the manuscript - lower right corner - not the journal PDF pages numbers)

Introduction:

The first two paragraphs had some overlapping points - e.g., the last sentence in each paragraph gets at the issue of looking at species whose mating system differs from close relatives (page 3, lines 6-13 and again lines 25-30). I think these paragraphs could be merged into a single paragraph.

The abbreviation "EPP" is given in the first paragraph (page 2, line 52), but is never defined (and while I know the definition, I think it should be given as well). Alternatively, since that is the only use I noted of EPP, it could be spelled out for that single use.

Page 4, line 45: The lack of lekking is described as a "loss", yet the justification is not provided. Elsewhere in the manuscript this is also referred to as a loss (or other phrases that indicate directionality of change). Providing some background or justification for these statements would strengthen those suggestions - what is known about the phylogeny of these taxa?

Similarly, clarifying the relationship to *Xenopipo* might also be good. Are these thought to be sister taxa? (suggesting possibly one loss of lekking behavior) or not?

Xenopipo is described as being largely monochromatic in the introduction, yet it is not until later in the manuscript that it is clarified that the Araripe manakin is not. Given that this also defines a difference between these two genera of non-lekking manakins, it could be helpful to readers to set this up earlier and clarify that the Araripe manakin is dichromatic here in the introduction.

Methods:

Page 7, lines 6-8: As written, this suggests that molecular sexing was used to separate adult versus sub-adult males, though I suspect it was that males were classified as adult if they had adult plumage, and sub-adult if they were green and the molecular sexing indicated they were males. This could be clarified.

Page 7j, line 36. I am not clear what is meant by "males remained in groups during the non-breeding season.". Since males were not in groups during the breeding season, does this mean that males formed multi-male groups in the non-breeding season (and presumably ceased being territorial)? This could be clarified.

Page 8, section d (parentage): Since assigning parentage is an important part of the conclusions, I think more details about the settings in CERVUS that were used for assignment should be provided in the main text and not supplementary material (a few sentences should provide critical details - I am find with details of the PCR and loci being provided only in supplementary). If space is limited, there are a lot of details of the field work and those may be able to be shortened slightly to allow for a few additional explanatory sentences regarding parentage assignment.

In addition, if the authors performed repeated genotyping of some or all individuals to check for consistency (I did not see this in the supplementary materials), a statement about this would also strengthen the results. (clarifying what was done if differences were identified).

Page 8, line 49: It reads "vary greatly" - can you clarify what varies greatly (among individuals? Among years? Etc.).

Results:

Page 9, line 26: For the 48 banded males that were resighted, what proportion was that of all banded males? Also, were these all males in adult plumage? You provide some of this information this this later in the results, but some reorganization might allow grouping of related information together in a way that may make this flow better.

Page 10, line 22: This first part of this paragraph describes observed mating behaviors, and the last half (beginning at the "However at line 22) begins the genetic details. I think it would highlight the genetic results more if this were separated into its own paragraph.

Page 10, line 34: For the male mating success, does the data included in figure 2 include data from multiple years for some males (e.g., did the male that sired 5 offspring do so in a single season, or across multiple seasons)?

Figure 2: I think this information could be presented more succinctly by using a histogram that reported the number of males siring 0, 1, 2, 3, or 5 offspring (smaller figure, and easier to grasp the patterns from it).

Discussion:

Page 12, line 8: The discussion states "...our study shows that one species of manakin has a mating system that is strongly divergent from that of its close relatives..." While the mating system of the Araripe manakin has some differences from typical manakins, one could argue it is still what might be considered a promiscuous system - albeit with lower male mating skews and higher rates of female multiple mating than in more traditional lekking manakins (later in the discussion, these points are raised that there are still some similarities with typical manakins). If the results had suggested a more typical socially monogamous mating system (clear pair-bonds, males siring many offspring in their territories, males provisioning nestlings), I would agree this system is strongly divergent. However, given the information, I would not classify this as "strongly divergent" from close relatives.

This first paragraph also makes the helmeted and Araripe manakin sound unique, but the introduction clarified that other species (*Xenopipo*) has presumably lost lekking behavior and has a more traditional monogamous mating system. Overall I was surprised the discussion did not reference *Xenopipo* at all (e.g., based on what is known about *Xenopipo*, how is it similar or different from the Araripe manakin?).

Page 12, line 10: There is no data presented on environmental pressures and how this might have shaped mating system changes (so I see no support for the statement "provides evidence that environmental pressures can play a strong role in shaping mating systems"). After finishing the manuscript I realize there is some speculation later in the discussion (after this initial statement) that describes how the environmental conditions may have affected mating system. I would recommend removing this statement from this part of the discussion.

Page 12, line 15: I am not clear what the "historical context" is that was provided, and think some brief expansion of the idea would help readers better understand the point being made.

Page 13, beginning line 11: Good points raised here, as this study definitely made me question the value of territories for this species (and whether females nested randomly with respect to male territories, or were attracted to territories - as is implied at the end of the discussion on page 14, lines 15-17).

Pages 13/14 (paragraph beginning "Both Araripe and helmeted..."): This paragraph raises some potential explanations, and I throw out a few other considerations for the authors: Some of what is written here might suggest bottlenecks in the recent past of this species and/or historically small populations - making drift a possible explanation for (at least some) differences. Is there any information on this? Given what is presented on the helmeted manakin, it is likely that this species may have a similar mating system as found in the Araripe manakin, then the shifts in behavior may have pre-dated the divergence between the two species, suggesting that is where selection (or drift) may have acted.

Review form: Reviewer 2 (David B. McDonald)

Is the manuscript scientifically sound in its present form?

Yes

Are the interpretations and conclusions justified by the results?

Yes

Is the language acceptable?

Yes

Do you have any ethical concerns with this paper?

No

Have you any concerns about statistical analyses in this paper?

No

Recommendation?

Accept with minor revision (please list in comments)

Comments to the Author(s)

This is an interesting and thought-provoking manuscript. The paper shows, convincingly, that the breeding biology of the Araripe manakin differs substantially from that of any other well-studied manakin; manakins are the "lekkiest" family of animals (McDonald 2010; perhaps the only family with a majority of lek-mating species, with the birds-of-paradise a distant second). Particularly interesting is the high rate of paternity for predefinitive males, suggesting that the striking definitive male plumage plays a role in territory acquisition but not in mate choice.

A major suggestion is to eliminate references to "adult," "sub-adult" and "juvenile" (which potentially confounds somatic and reproductive maturation), and instead use "definitive" and "predefinitive" as proposed by Foster (1987). Clearly the predefinitive males are reproductively mature, as shown by the very surprising results of the parentage analyses. If you have any idea of age of acquisition of definitive plumage, that would be very interesting in terms of the range of delays in Chiroxiphia (Doucet et al. 2007). Reading between the lines, I suspect that males sometimes (always?) delay plumage maturation until two years after hatch. If you have the data, note that explicitly.

A second suggestion is to note that *Antilophia* is "sister" to *Chiroxiphia* (Ohlson et al. 2013; also Hackett et al. and Anciães and Peterson). *Chiroxiphia* (especially *C. linearis*) is perhaps the extreme in lek-mating systems, with the variance in mating success so high that it has driven the unique feature of obligate dual-male courtship. That these two sister genera have such divergent mating systems demonstrates forcefully that ecology can outweigh phylogeny in shaping mating systems (Emlen and Oring 1977, though cited elsewhere in the paper, could be singled out here). *Antilophia* (p. 13, l. 45) and *Chiroxiphia* also share an unusual habitat type (gallery forest) that is more xeric than that of virtually any other manakin genus.

I suggest broadening the generality concerning the potential flexibility of lek mating systems. It is worth mentioning the work of Thirgood et al. (1999) on fallow deer and that of Cartar and Lyon (1988) on Buff-breasted Sandpipers (but see Lanctot and Weatherhead 1997). Perhaps also refer again to Emlen and Oring's (1977) fundamental proposition that ecology drives mating systems.

More minor suggestions include:

Abstract l. 26. "Territorial males sired only 7% of the nestlings from the nests inside their territories." [not the very vague "most"].

Abstract, l. 28 "socially monogamous" [as correctly used elsewhere]

p. 3, l. 35. "various" not "different"

p. 6, l. 55 and throughout. "predefinitive", "definitive" not "adult", "sub-adult", "juvenile."

p. 12, l. 47. "Finally, the percentage (48%) of nestlings sired by predefinitive males is noteworthy." [Mean age of copulators in *C. linearis* is 10.1 years!].

p. 12, l. 52. "clustered" not "cooperative." I see no real evidence that the clustered displays are cooperative. Indeed, the chasing perhaps suggests agonism. But the clustering is an interesting similarity to lek arenas.

Territory boundaries: Were the territories delimited as minimum convex polygons or by adding an (unmentioned) buffer around the singing perches? That matters to the figure (and to an understanding of how male territories function).

Figure 2. It would be informative to use three types of shading for the bars: predefinitive males, territorial males, floater definitive males.

Literature cited:

Cartar, R.V., and B.E. Lyon. 1988. The mating system of the Buff-breasted Sandpiper: lekking and resource defense polygyny. *Ornis Scandinavica* 19: 74-76.

- Doucet, S.M., D.B. McDonald, M.S. Foster, and R.P. Clay. 2007. Plumage development and molt in Long-tailed Manakins (*Chiroxiphia linearis*): variation according to sex and age. *Auk* 124: 29-43.
- Foster, M.S. 1987. Delayed maturation, neoteny, and social system differences in two manakins of the genus *Chiroxiphia*. *Evol.* 41: 547-58.
- Lanctot, R.B. and P.J. Weatherhead. 1997. Ephemeral lekking behavior in the buff-breasted sandpiper, *Tryngites subruficollis*. *Behav. Ecol.* 8: 268-278.
- McDonald, D.B. 2010. A spatial dance to the music of time in the leks of long-tailed manakins. *Advances in the Study of Behavior* 42: 55-81. Elsevier, 2010. [https://doi.org/10.1016/S0065-3454\(10\)42002-1](https://doi.org/10.1016/S0065-3454(10)42002-1).
- Thirgood, S.J., J. Langbein, and R.J. Putnam. 1999. Intraspecific variation in ungulate mating strategies: The case of the flexible fallow deer. *Adv. Study Behav.* 28: 333-361.
- Ohlson, Jan I., Jon Fjeldså, and Per G.P. Ericson. 2013. Molecular Phylogeny of the Manakins (Aves: Passeriformes: Pipridae), with a New Classification and the Description of a New Genus. *Molecular Phylogenetics and Evolution* 69: 796-804. <https://doi.org/10.1016/j.ympev.2013.06.024>.

Decision letter (RSOS-191548.R0)

04-Nov-2019

Dear Dr Macedo

On behalf of the Editors, I am pleased to inform you that your Manuscript RSOS-191548 entitled "An Atypical Mating System in a Neotropical Manakin" has been accepted for publication in Royal Society Open Science subject to minor revision in accordance with the referee suggestions. Please find the referees' comments at the end of this email.

The reviewers and handling editors have recommended publication, but also suggest some minor revisions to your manuscript. Therefore, I invite you to respond to the comments and revise your manuscript.

- Ethics statement

- Data accessibility

If you wish to submit your supporting data or code to Dryad (<http://datadryad.org/>), or modify your current submission to dryad, please use the following link:
<http://datadryad.org/submit?journalID=RSOS&manu=RSOS-191548>

- **Competing interests**

- **Authors' contributions**

- **Acknowledgements**

- **Funding statement**

Because the schedule for publication is very tight, it is a condition of publication that you submit the revised version of your manuscript before 13-Nov-2019. Please note that the revision deadline will expire at 00.00am on this date. If you do not think you will be able to meet this date please let me know immediately.

When submitting your revised manuscript, you will be able to respond to the comments made by the referees and upload a file "Response to Referees" in "Section 6 - File Upload". You can use this to document any changes you make to the original manuscript. In order to expedite the

processing of the revised manuscript, please be as specific as possible in your response to the referees. We strongly recommend uploading two versions of your revised manuscript:

Kind regards,
Andrew Dunn
Senior Publishing Editor
Royal Society Open Science Editorial Office

on behalf of Dr Alexander Ophir (Associate Editor) and Kevin Padian (Subject Editor)
openscience@royalsociety.org

Associate Editor Comments to Author (Dr Alexander Ophir):

Dear Dr. Macedo,

I have received reviews from two leading experts in the field, who were both very positive about your paper. I concur with their positive assessment of your manuscript and believe that this will be a very nice addition to the manakin, lek, and mating system literature that is sure to have broad appeal. Both reviewers provided a list of helpful and fairly minor points, that your manuscript will benefit from if you can integrate or address. I would like to congratulate you and your co-authors on another excellent paper.

Best
Alex Ophir
Associate Editor, RSOS

Reviewer comments to Author:

Reviewer: 1

Comments to the Author(s)

This paper describes the intriguing mating system of the Araripe manakin. Unlike more typical manakins that form leks where males have high mating skew, in this species males defend a more tradition territory and the skew in male mating success is much lower. Females of other manakins have relatively low levels of broods sired by multiple males, this may be more common in the Araripe manakin. However, unlike more traditional (socially) monogamous systems where males may sire many of the offspring raised in nests within their territory and provision offspring, males in the Araripe manakin rarely sired chicks in nests within their territories and never provisioned offspring. In addition, territory holding does not appear necessary for males to reproduce - nearly half of the offspring were sired by males that were not yet in adult plumage (and these green males were not observed defending territories). Thus this system is not quite like a typical manakin (though elements of that are present), though it is also not like a typical socially monogamous passerine.

While the overall study is well-done, and the presentation was good, there are a number of areas where I felt there should be some additional clarification or details provided to strengthen the manuscript.

Specific Comments (Page numbers refer to the manuscript - lower right corner - not the journal PDF pages numbers)

Introduction:

The first two paragraphs had some overlapping points - e.g., the last sentence in each paragraph gets at the issue of looking at species whose mating system differs from close relatives (page 3, lines 6-13 and again lines 25-30). I think these paragraphs could be merged into a single paragraph.

The abbreviation "EPP" is given in the first paragraph (page 2, line 52), but is never defined (and while I know the definition, I think it should be given as well). Alternatively, since that is the only use I noted of EPP, it could be spelled out for that single use.

Page 4, line 45: The lack of lekking is described as a "loss", yet the justification is not provided. Elsewhere in the manuscript this is also referred to as a loss (or other phrases that indicate directionality of change). Providing some background or justification for these statements would strengthen those suggestions - what is known about the phylogeny of these taxa?

Similarly, clarifying the relationship to *Xenopipo* might also be good. Are these thought to be sister taxa? (suggesting possibly one loss of lekking behavior) or not?

Xenopipo is described as being largely monochromatic in the introduction, yet it is not until later in the manuscript that it is clarified that the *Araripe* manakin is not. Given that this also defines a difference between these two genera of non-lekking manakins, it could be helpful to readers to set this up earlier and clarify that the *Araripe* manakin is dichromatic here in the introduction.

Methods:

Page 7, lines 6-8: As written, this suggests that molecular sexing was used to separate adult versus sub-adult males, though I suspect it was that males were classified as adult if they had adult plumage, and sub-adult if they were green and the molecular sexing indicated they were males. This could be clarified.

Page 7j, line 36. I am not clear what is meant by "males remained in groups during the non-breeding season.". Since males were not in groups during the breeding season, does this mean that males formed multi-male groups in the non-breeding season (and presumably ceased being territorial)? This could be clarified.

Page 8, section d (parentage): Since assigning parentage is an important part of the conclusions, I think more details about the settings in CERVUS that were used for assignment should be provided in the main text and not supplementary material (a few sentences should provide critical details - I am find with details of the PCR and loci being provided only in supplementary). If space is limited, there are a lot of details of the field work and those may be able to be shortened slightly to allow for a few additional explanatory sentences regarding parentage assignment.

In addition, if the authors performed repeated genotyping of some or all individuals to check for consistency (I did not see this in the supplementary materials), a statement about this would also strengthen the results. (clarifying what was done if differences were identified).

Page 8, line 49: It reads "vary greatly" - can you clarify what varies greatly (among individuals? Among years? Etc.).

Results:

Page 9, line 26: For the 48 banded males that were resighted, what proportion was that of all banded males? Also, were these all males in adult plumage? You provide some of this information this this later in the results, but some reorganization might allow grouping of related information together in a way that may make this flow better.

Page 10, line 22: This first part of this paragraph describes observed mating behaviors, and the

last half (beginning at the "However at line 22) begins the genetic details. I think it would highlight the genetic results more if this were separated into its own paragraph.

Page 10, line 34: For the male mating success, does the data included in figure 2 include data from multiple years for some males (e.g., did the male that sired 5 offspring do so in a single season, or across multiple seasons)?

Figure 2: I think this information could be presented more succinctly by using a histogram that reported the number of males siring 0, 1, 2, 3, or 5 offspring (smaller figure, and easier to grasp the patterns from it).

Discussion:

Page 12, line 8: The discussion states "...our study shows that one species of manakin has a mating system that is strongly divergent from that of its close relatives..." While the mating system of the Araripe manakin has some differences from typical manakins, one could argue it is still what might be considered a promiscuous system - albeit with lower male mating skews and higher rates of female multiple mating than in more traditional lekking manakins (later in the discussion, these points are raised that there are still some similarities with typical manakins). If the results had suggested a more typical socially monogamous mating system (clear pair-bonds, males siring many offspring in their territories, males provisioning nestlings), I would agree this system is strongly divergent. However, given the information, I would not classify this as "strongly divergent" from close relatives.

This first paragraph also makes the helmeted and Araripe manakin sound unique, but the introduction clarified that other species (*Xenopipo*) has presumably lost lekking behavior and has a more traditional monogamous mating system. Overall I was surprised the discussion did not reference *Xenopipo* at all (e.g., based on what is known about *Xenopipo*, how is it similar or different from the Araripe manakin?).

Page 12, line 10: There is no data presented on environmental pressures and how this might have shaped mating system changes (so I see no support for the statement "provides evidence that environmental pressures can play a strong role in shaping mating systems"). After finishing the manuscript I realize there is some speculation later in the discussion (after this initial statement) that describes how the environmental conditions may have affected mating system. I would recommend removing this statement from this part of the discussion.

Page 12, line 15: I am not clear what the "historical context" is that was provided, and think some brief expansion of the idea would help readers better understand the point being made.

Page 13, beginning line 11: Good points raised here, as this study definitely made me question the value of territories for this species (and whether females nested randomly with respect to male territories, or were attracted to territories - as is implied at the end of the discussion on page 14, lines 15-17).

Pages 13/14 (paragraph beginning "Both Araripe and helmeted..."): This paragraph raises some potential explanations, and I throw out a few other considerations for the authors: Some of what is written here might suggest bottlenecks in the recent past of this species and/or historically small populations - making drift a possible explanation for (at least some) differences. Is there any information on this? Given what is presented on the helmeted manakin, it is likely that this species may have a similar mating system as found in the Araripe manakin, then the shifts in behavior may have pre-dated the divergence between the two species, suggesting that is where selection (or drift) may have acted.

Reviewer: 2

Comments to the Author(s)

This is an interesting and thought-provoking manuscript. The paper shows, convincingly, that the breeding biology of the Araripe manakin differs substantially from that of any other well-studied manakin; manakins are the "lekkiest" family of animals (McDonald 2010; perhaps the only family with a majority of lek-mating species, with the birds-of-paradise a distant second). Particularly interesting is the high rate of paternity for predefinitive males, suggesting that the striking definitive male plumage plays a role in territory acquisition but not in mate choice.

A major suggestion is to eliminate references to "adult," "sub-adult" and "juvenile" (which potentially confounds somatic and reproductive maturation), and instead use "definitive" and "predefinitive" as proposed by Foster (1987). Clearly the predefinitive males are reproductively mature, as shown by the very surprising results of the parentage analyses. If you have any idea of age of acquisition of definitive plumage, that would be very interesting in terms of the range of delays in *Chiroxiphia* (Doucet et al. 2007). Reading between the lines, I suspect that males sometimes (always?) delay plumage maturation until two years after hatch. If you have the data, note that explicitly.

A second suggestion is to note that *Antilophia* is "sister" to *Chiroxiphia* (Ohlson et al. 2013; also Hackett et al. and Anciães and Peterson). *Chiroxiphia* (especially *C. linearis*) is perhaps the extreme in lek-mating systems, with the variance in mating success so high that it has driven the unique feature of obligate dual-male courtship. That these two sister genera have such divergent mating systems demonstrates forcefully that ecology can outweigh phylogeny in shaping mating systems (Emlen and Oring 1977, though cited elsewhere in the paper, could be singled out here). *Antilophia* (p. 13, l. 45) and *Chiroxiphia* also share an unusual habitat type (gallery forest) that is more xeric than that of virtually any other manakin genus.

I suggest broadening the generality concerning the potential flexibility of lek mating systems. It is worth mentioning the work of Thirgood et al. (1999) on fallow deer and that of Cartar and Lyon (1988) on Buff-breasted Sandpipers (but see Lanctot and Weatherhead 1997). Perhaps also refer again to Emlen and Oring's (1977) fundamental proposition that ecology drives mating systems.

More minor suggestions include:

Abstract l. 26. "Territorial males sired only 7% of the nestlings from the nests inside their territories." [not the very vague "most"].

Abstract, l. 28 "socially monogamous" [as correctly used elsewhere]

p. 3, l. 35. "various" not "different"

p. 6, l. 55 and throughout. "predefinitive", "definitive" not "adult", "sub-adult", "juvenile."

p. 12, l. 47. "Finally, the percentage (48%) of nestlings sired by predefinitive males is noteworthy." [Mean age of copulators in *C. linearis* is 10.1 years!].

p. 12, l. 52. "clustered" not "cooperative." I see no real evidence that the clustered displays are cooperative. Indeed, the chasing perhaps suggests agonism. But the clustering is an interesting similarity to lek arenas.

Territory boundaries: Were the territories delimited as minimum convex polygons or by adding an (unmentioned) buffer around the singing perches? That matters to the figure (and to an understanding of how male territories function).

Figure 2. It would be informative to use three types of shading for the bars: predefinitive males, territorial males, floater definitive males.

Literature cited:

- Cartar, R.V., and B.E. Lyon. 1988. The mating system of the Buff-breasted Sandpiper: lekking and resource defense polygyny. *Ornis Scandinavica* 19: 74-76.
- Doucet, S.M., D.B. McDonald, M.S. Foster, and R.P. Clay. 2007. Plumage development and molt in Long-tailed Manakins (*Chiroxiphia linearis*): variation according to sex and age. *Auk* 124: 29-43.
- Foster, M.S. 1987. Delayed maturation, neoteny, and social system differences in two manakins of the genus *Chiroxiphia*. *Evol.* 41: 547-58.
- Lanctot, R.B. and P.J. Weatherhead. 1997. Ephemeral lekking behavior in the buff-breasted sandpiper, *Tryngites subruficollis*. *Behav. Ecol.* 8: 268-278.
- McDonald, D.B. 2010. A spatial dance to the music of time in the leks of long-tailed manakins. *Advances in the Study of Behavior* 42: 55-81. Elsevier, 2010. [https://doi.org/10.1016/S0065-3454\(10\)42002-1](https://doi.org/10.1016/S0065-3454(10)42002-1).
- Thirgood, S.J., J. Langbein, and R.J. Putnam. 1999. Intraspecific variation in ungulate mating strategies: The case of the flexible fallow deer. *Adv. Study Behav.* 28: 333-361.
- Ohlson, Jan I., Jon Fjeldså, and Per G.P. Ericson. 2013. Molecular Phylogeny of the Manakins (Aves: Passeriformes: Pipridae), with a New Classification and the Description of a New Genus. *Molecular Phylogenetics and Evolution* 69: 796-804. <https://doi.org/10.1016/j.ympev.2013.06.024>.

Author's Response to Decision Letter for (RSOS-191548.R0)

See Appendix A.

Decision letter (RSOS-191548.R1)

17-Nov-2019

Dear Dr Macedo,

It is a pleasure to accept your manuscript entitled "An Atypical Mating System in a Neotropical Manakin" in its current form for publication in Royal Society Open Science. The comments of the reviewer(s) who reviewed your manuscript are included at the foot of this letter.

on behalf of Dr Alexander Ophir (Associate Editor) and Kevin Padian (Subject Editor)
openscience@royalsociety.org

Appendix A

November 12, 2019

Dr. Andrew Dunn, Senior Publishing Editor
Dr. Alex Ophir, Associate Editor
Royal Society Open Science

Ref: Manuscript ID RSOS-191548

An Atypical Mating System in a Neotropical Manakin

Dear Drs. Dunn and Ophir:

We are very pleased that our manuscript was accepted for publication in Royal Society Open Science with only minor revisions. We found the comments and suggestions of the reviewers relevant for improving the clarity and importance of our study. We have now revised the manuscript, and addressed all matters of concern appropriately. Please find our comments below, **in boldface**, in response to the suggestions made by the two reviewers. We are submitting a highlighted version of the manuscript so that the changes made can be easily tracked, as well as a “clean” copy without the highlighted markings.

Sincerely,

Regina H. Macedo, in behalf of all authors

General editorial instructions

Please ensure you have prepared your revision in accordance with the guidance at <https://royalsociety.org/journals/authors/author-guidelines/> -- please note that we cannot publish your manuscript without the end statements.

In our previous version of the manuscript we already had the end statements in place, including the Dryad Digital Repository link to our data. We've made slight changes to some of the sections based on the model provided by the journal. Changes are highlighted in the section of the end statements (page 16).

Reviewer: 1

Introduction:

The first two paragraphs had some overlapping points - e.g., the last sentence in each paragraph gets at the issue of looking at species whose mating system differs from close relatives (page 3, lines 6-13 and again lines 25-30). I think these paragraphs could be merged into a single paragraph.

Done. We have moved the overlapping information to the end of a single merged paragraph (page 3, lines 6-12).

The abbreviation "EPP" is given in the first paragraph (page 2, line 52), but is never defined (and while I know the definition, I think it should be given as well). Alternatively, since that is the only use I noted of EPP, it could be spelled out for that single use.

Since it's the only mention of extrapair paternity, we decided to spell it out and avoid the acronym (page 2, first sentence of Introduction).

Page 4, line 45: The lack of lekking is described as a "loss", yet the justification is not provided. Elsewhere in the manuscript this is also referred to as a loss (or other phrases that indicate directionality of change). Providing some background or justification for these statements would strengthen those suggestions - what is known about the phylogeny of these taxa?

A social/behavioral phylogeny of the manakins was provided by Prum (1994), based on the syringeal hypothesis of phylogeny (Prom 1992). Subsequent molecular phylogenies do not contradict the proposal of a single origin in the monophyletic clade of manakins. We have added a small section of text to highlight this information to allow us to subsequently conclude that in some manakin species, the lack of lekking can be considered a "loss". (Page 4, lines 14-15: paragraph initiates with "Lekking is considered an ancestral trait...")

Similarly, clarifying the relationship to *Xenopipo* might also be good. Are these thought to be sister taxa? (suggesting possibly one loss of lekking behavior) or not?

***Xenopipo*, at the time of Prum's behavioral analysis, was behaviorally unknown, but suspected of being non-lekking. This was later verified in a behavioral study of two**

species in the *Xenopipo* genus (Ribeiro et al. citation). The two genera (*Xenopipo* and *Antilophia*) are distantly related, though, so a discussion of the phylogenetic finer distinctions didn't necessary to illustrate the point we were making, i.e., that few species in the clade deviate from lek mating.

Xenopipo is described as being largely monochromatic in the introduction, yet it is not until later in the manuscript that it is clarified that the Araripe manakin is not. Given that this also defines a difference between these two genera of non-lekking manakins, it could be helpful to readers to set this up earlier and clarify that the Araripe manakin is dichromatic here in the introduction.

We have added the information that both the helmeted and Araripe manakins are highly dichromatic (page 4 line 19; and page 5 lines 5-6)

Methods:

Page 7, lines 6-8: As written, this suggests that molecular sexing was used to separate adult versus sub-adult males, though I suspect it was that males were classified as adult if they had adult plumage, and sub-adult if they were green and the molecular sexing indicated they were males. This could be clarified.

We switched the order and merged the two sentences, so that the description about how to discern male adults and sub-adults based on plumage comes first, which hopefully has clarified this point. (page 7, lines 6-8). Note that in response to a suggestion from Reviewer 2, we have now substituted all references of adult and subadult males with definitive and predefinitive plumage descriptors.

Page 7j, line 36. I am not clear what is meant by "males remained in groups during the non-breeding season.". Since males were not in groups during the breeding season, does this mean that males formed multi-male groups in the non-breeding season (and presumably ceased being territorial)? This could be clarified.

This is exactly what happens in the non-breeding season—males become nonterritorial and form multi-male aggregations. We've clarified this point (page 7 lines 20-21, last paragraph).

Page 8, section d (parentage): Since assigning parentage is an important part of the conclusions, I think more details about the settings in CERVUS that were used for assignment should be provided in the main text and not supplementary material (a few sentences should provide critical details - I am find with details of the PCR and loci being provided only in supplementary). If space is limited, there are a lot of details of the field

work and those may be able to be shortened slightly to allow for a few additional explanatory sentences regarding parentage assignment.

In addition, if the authors performed repeated genotyping of some or all individuals to check for consistency (I did not see this in the supplementary materials), a statement about this would also strengthen the results. (clarifying what was done if differences were identified).

We have included in the text additional details about the CERVUS analyses, replicated from the Supplementary Material (page 8 lines 15-21 and page 9 lines 1-2).

Page 8, line 49: It reads "vary greatly" - can you clarify what varies greatly (among individuals? Among years? Etc.).

We meant that reproductive success did not vary greatly among individuals. We've added this to the sentence to clarify (page 9 line 10).

Results:

Page 9, line 26: For the 48 banded males that were resighted, what proportion was that of all banded males? Also, were these all males in adult plumage? You provide some of this information this this later in the results, but some reorganization might allow grouping of related information together in a way that may make this flow better.

We reorganized the order of presentation in this part of the text, which we think has improved the flow of information. We also added the details that the defended territories were held by males in definitive plumage (previously referred to as "adult plumage"). (page 10, lines 2-8)

Page 10, line 22: This first part of this paragraph describes observed mating behaviors, and the last half (beginning at the "However at line 22) begins the genetic details. I think it would highlight the genetic results more if this were separated into its own paragraph.

Done—we started the genetic information in a new paragraph. (page 11 lines 3-5).

Page 10, line 34: For the male mating success, does the data included in figure 2 include data from multiple years for some males (e.g., did the male that sired 5 offspring do so in a single season, or across multiple seasons)?

We clarified this by changing the sentence so that it reads: “The distribution of siring success across males sampled in three breeding seasons ...” (page 11, lines 8-9). This information had already been included in the legend of Figure 2.

Figure 2: I think this information could be presented more succinctly by using a histogram that reported the number of males siring 0, 1, 2, 3, or 5 offspring (smaller figure, and easier to grasp the patterns from it).

We accepted this suggestion and modified Figure 2 using a histogram showing the number of males that sired different numbers of offspring.

Discussion:

Page 12, line 8: The discussion states "...our study shows that one species of manakin has a mating system that is strongly divergent from that of its close relatives..." While the mating system of the Araripe manakin has some differences from typical manakins, one could argue it is still what might be considered a promiscuous system - albeit with lower male mating skews and higher rates of female multiple mating than in more traditional lekking manakins (later in the discussion, these points are raised that there are still some similarities with typical manakins). If the results had suggested a more typical socially monogamous mating system (clear pair-bonds, males siring many offspring in their territories, males provisioning nestlings), I would agree this system is strongly divergent. However, given the information, I would not classify this as "strongly divergent" from close relatives.

We changed the text in this section to be more specific so that we are not asserting that the mating system as a whole is strongly divergent. Instead, we now have the text stating that “...the Araripe manakin has a mating system that differs substantially in many characteristics from that of its close relatives...” (page 11 line 17).

This first paragraph also makes the helmeted and Araripe manakin sound unique, but the introduction clarified that other species (*Xenopipo*) has presumably lost lekking behavior and has a more traditional monogamous mating system. Overall I was surprised the discussion did not reference *Xenopipo* at all (e.g., based on what is known about *Xenopipo*, how is it similar or different from the Araripe manakin?).

We did not provide more details about *Xenopipo* because of the restricted length limitation of the manuscript. The *Xenopipo* study was cited in the Introduction because the genus also contains species that differ from the majority of other piprids

in some aspects of morphology and behavior. The study was based on plumage and size traits showing that it is a monochromatic species and also indicated the absence of lekking behavior. However, that study did not include genetic analyses, so it differs in what it can conclude relative to the genetic mating system in comparison to what we focus on in our Discussion, which is the dissociation between male territoriality and genetic paternity. We didn't feel that we should focus on *Xenopipo* in our Discussion because it is phylogenetically distant from *Antilophia*, and other selective pressures may be involved.

Page 12, line 10: There is no data presented on environmental pressures and how this might have shaped mating system changes (so I see no support for the statement "provides evidence that environmental pressures can play a strong role in shaping mating systems"). After finishing the manuscript I realize there is some speculation later in the discussion (after this initial statement) that describes how the environmental conditions may have affected mating system. I would recommend removing this statement from this part of the discussion.

We agree and have deleted this part of the sentence from the text (Page 12, text removed from the 3rd line in the Discussion).

Page 12, line 15: I am not clear what the "historical context" is that was provided, and think some brief expansion of the idea would help readers better understand the point being made.

We changed this sentence to exclude the wording "historical context", which is probably what was confusing the issue, and inserted enough details to contextualize the rest of the information (page 12, lines 18-19).

Page 13, beginning line 11: Good points raised here, as this study definitely made me question the value of territories for this species (and whether females nested randomly with respect to male territories, or were attracted to territories - as is implied at the end of the discussion on page 14, lines 15-17).

Indeed! We were very puzzled and it was difficult to provide alternative explanations for territoriality in the species.

Pages 13/14 (paragraph beginning "Both Araripe and helmeted...): This paragraph raises some potential explanations, and I throw out a few other considerations for the authors: Some of what is written here might suggest bottlenecks in the recent past of this species and/or historically small populations - making drift a possible explanation

for (at least some) differences. Is there any information on this? Given what is presented on the helmeted manakin, it is likely that this species may have a similar mating system as found in the Araripe manakin, then the shifts in behavior may have pre-dated the divergence between the two species, suggesting that is where selection (or drift) may have acted.

We don't have other information to substantiate these speculations (unfortunately). Nonetheless, we've added the bottleneck and drift possibility to our discussion. We are presently conducting a genetic and behavioral study of the helmeted manakin to assess the similarities in genetic mating system between the two species! Hopefully, we will be able to provide a comparative analysis for the two species in the near future. (pages 14, lines 18-20, starting with "Among other possible mechanisms...").

Reviewer: 2

Comments to the Author(s)

This is an interesting and thought-provoking manuscript. The paper shows, convincingly, that the breeding biology of the Araripe manakin differs substantially from that of any other well-studied manakin; manakins are the "lekkiest" family of animals (McDonald 2010; perhaps the only family with a majority of lek-mating species, with the birds-of-paradise a distant second). Particularly interesting is the high rate of paternity for predefinitive males, suggesting that the striking definitive male plumage plays a role in territory acquisition but not in mate choice.

A major suggestion is to eliminate references to "adult," "sub-adult" and "juvenile" (which potentially confounds somatic and reproductive maturation), and instead use "definitive" and "predefinitive" as proposed by Foster (1987). Clearly the predefinitive males are reproductively mature, as shown by the very surprising results of the parentage analyses. If you have any idea of age of acquisition of definitive plumage, that would be very interesting in terms of the range of delays in *Chiroxiphia* (Doucet et al. 2007). Reading between the lines, I suspect that males sometimes (always?) delay plumage maturation until two years after hatch. If you have the data, note that explicitly.

We've followed this suggestion, of using "predefinitive" and "definitive" throughout the manuscript (all changes highlighted). The information that we have about plumage maturation is from a single offspring that we banded in the nest and that was recaptured two years later, and this indeed shows that they can delay maturation of

the plumage for, at the very least, two years. We made note of this information (page 7, lines 7-9).

A second suggestion is to note that *Antilophia* is "sister" to *Chiroxiphia* (Ohlson et al. 2013; also Hackett et al. and Anciães and Peterson). *Chiroxiphia* (especially *C. linearis*) is perhaps the extreme in lek-mating systems, with the variance in mating success so high that it has driven the unique feature of obligate dual-male courtship. That these two sister genera have such divergent mating systems demonstrates forcefully that ecology can outweigh phylogeny in shaping mating systems (Emlen and Oring 1977, though cited elsewhere in the paper, could be singled out here). *Antilophia* (p. 13, l. 45) and *Chiroxiphia* also share an unusual habitat type (gallery forest) that is more xeric than that of virtually any other manakin genus.

We have included the perspectives suggested by the reviewer including some of his/her specific wording. We chose to include the Ohlson et al. (2013) reference and one other we found relevant for this point, which is Silva et al. (2018). We also cited Emlen and Oring (1977) again, as suggested (page 14, lines 14-18).

I suggest broadening the generality concerning the potential flexibility of lek mating systems. It is worth mentioning the work of Thirgood et al. (1999) on fallow deer and that of Cartar and Lyon (1988) on Buff-breasted Sandpipers (but see Lanctot and Weatherhead 1997). Perhaps also refer again to Emlen and Oring's (1977) fundamental proposition that ecology drives mating systems.

We avoided including the examples provided above (sandpipers and fallow deer) because they refer to intraspecific flexibility in mating system, wherein some individuals or populations are lek mating while others are resource-driven polygynies. Our discussion focuses on inter-specific variation within the clade, so we think it would only confuse our discussion if we brought in this additional perspective, without the possibility of developing it more in-depth. On the other hand, we referred back to Emlen and Oring's (1977) classical paper (page 15, lines 11-129).

More minor suggestions include:

Abstract l. 26. "Territorial males sired only 7% of the nestlings from the nests inside their territories." [not the very vague "most"].

Changed as suggested (Abstract: lines 8-9).

Abstract, l. 28 "socially monogamous" [as correctly used elsewhere] p. 3, l. 35. "various" not "different"

Changed “monogamous” to “socially monogamous” (Abstract, line 12).

Changed “different” to “various” (page 3, line 15)

p. 6, l. 55 and throughout. "predefinitive", "definitive" not "adult", "sub-adult", "juvenile."

Done here and throughout paper.

p. 12, l. 52. "clustered" not "cooperative." I see no real evidence that the clustered displays are cooperative. Indeed, the chasing perhaps suggests agonism. But the clustering is an interesting similarity to lek arenas.

Done. Changed “cooperative” to “clustered” (page 13, line 13)

Territory boundaries: Were the territories delimited as minimum convex polygons or by adding an (unmentioned) buffer around the singing perches? That matters to the figure (and to an understanding of how male territories function).

The territories were delimited based on the singing perches, as minimum convex polygons without adding buffers around perches.

Figure 2. It would be informative to use three types of shading for the bars: predefinitive males, territorial males, floater definitive males.

We did not take up this suggestion because we felt it would complicate the message intended by the figure, which in this case is simply the low variation in reproductive success among all breeding males. Additionally, the figure is now in the form of a histogram, in response to a suggestion made by Reviewer 1.